# Temporal Robustness against Data Poisoning

**Wenxiao Wang**
Department of Computer Science
University of Maryland
College Park, MD 20742
wwx@umd.edu

**Soheil Feizi**
Department of Computer Science
University of Maryland
College Park, MD 20742
sfeizi@umd.edu

## Abstract

Data poisoning considers cases when an adversary manipulates the behavior of machine learning algorithms through malicious training data. Existing threat models of data poisoning center around a single metric, the number of poisoned samples. In consequence, if attackers can poison more samples than expected with affordable overhead, as in many practical scenarios, they may be able to render existing defenses ineffective in a short time. To address this issue, we leverage timestamps denoting the birth dates of data, which are often available but neglected in the past. Benefiting from these timestamps, we propose a temporal threat model of data poisoning with two novel metrics, earliness and duration, which respectively measure how long an attack started in advance and how long an attack lasted. Using these metrics, we define the notions of temporal robustness against data poisoning, providing a meaningful sense of protection even with unbounded amounts of poisoned samples when the attacks are temporally bounded. We present a benchmark with an evaluation protocol simulating continuous data collection and periodic deployments of updated models, thus enabling empirical evaluation of temporal robustness. Lastly, we develop and also empirically verify a baseline defense, namely temporal aggregation, offering provable temporal robustness and highlighting the potential of our temporal threat model for data poisoning.

## 1 Introduction

Many applications of machine learning rely on training data collected from potentially malicious sources, which motivates the study of data poisoning. Data poisoning postulates an adversary who manipulates the behavior of machine learning algorithms [11] through malicious training data.

A great variety of poisoning attacks have been proposed to challenge the reliability of machine learning with unreliable data sources, including triggerless attacks [2, 22, 15, 30, 46, 1, 10] and backdoor attacks [8, 35, 29, 43, 32, 45], depending on whether input modifications are required at inference time. At the same time, diverse defenses are proposed, some of which aim at detecting/filtering poisoned samples [33, 9, 25, 34, 37, 23, 44], some of which aim at robustifying models trained with poisoned samples [41, 36, 27, 19, 12, 16, 13, 14, 6, 42, 7, 39, 26, 38, 40].

The number of poisoned samples is absolutely a key metric among existing threat models of data poisoning, and defenses are typically designed for robustness against bounded numbers of poisoned samples. However, the number of poisoned samples defenses may tolerate can be quite limited (e.g. $\leq 0.1\%$ of the dataset size), as supported empirically by [3, 4, 32, 10, 5] and theoretically by [40]. Furthermore, an explicit bound on the number of poisoned samples is not available in many practical scenarios and therefore it can be possible for attacks to render existing defenses ineffective in a short time. For example, when data are scraped from the internet, an attacker can always post more poisoned samples online; when data are contributed by users, a single attacker may pose itself as

37th Conference on Neural Information Processing Systems (NeurIPS 2023).

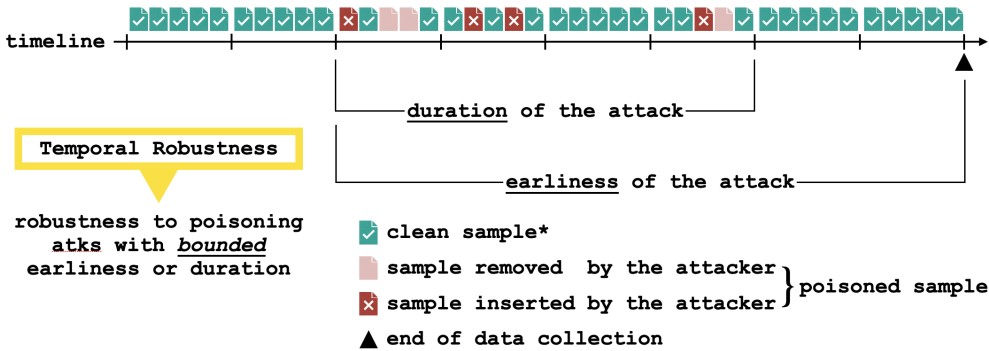

Figure 1: An illustration of our temporal modeling of data poisoning. Every sample, no matter whether it is clean or poisoned, is associated with a birth date, i.e. a timestamp denoting when the sample becomes available for collection. Two temporal metrics are proposed to characterize a data poisoning attack: **Earliness**, how long the poisoning attack started in advance, is determined by the earliest timestamp of all poisoned samples; **Duration**, how long the poisoning attack lasted, is defined as the time passed from the earliest to the latest timestamps of all poisoned samples.

multiple users to poison a lot more training samples. In these cases, the robustness notions in existing threat models of data poisoning fail to provide a meaningful sense of protection.

Is there any option other than the number of poisoned samples? Yes! When collecting data, there are usually corresponding timestamps available but neglected previously in modeling data poisoning. For instance, a company can keep track of the actual creation time of user data; online contents often come with a publication date indicating when they are posted. While attackers may create poisoned samples of many kinds, they typically have no direct way to set the values of these timestamps. In addition, temporally bounded poisoning attacks have been proposed in a couple specific scenarios [28, 5]. These motivate us to use time as a new dimension for defenses against data poisoning.

To summarize, we make the following contributions:

- proposing a **novel threat model** for data poisoning along with the notions of **temporal robustness**, providing meaningful protections even with potentially unbounded amounts of poisoned samples when the attacks are temporally bounded;

- designing a **benchmark** with an evaluation protocol that simulates continuous data collection with periodic model updates, enabling empirical evaluation of temporal robustness;

- developing and verifying a **baseline defense** with **provable** temporal robustness.

## 2 Temporal Modeling of Data Poisoning

### 2.1 Background: Threat Models that Center Around the Number of Poisoned Samples

Here we recap a typical threat model depending heavily on the number of poisoned samples.

Let $\mathcal{X}$ be the input space, $\mathcal{Y}$ be the label space and $\mathcal{D} = (\mathcal{X} \times \mathcal{Y})^{\mathbb{N}}$ be the space of all possible training sets. Let $f : \mathcal{D} \times \mathcal{X} \to \mathcal{Y}$ be a learner that maps an input to a label conditioning on a training set. Given a training set $D \in \mathcal{D}$, a learner $f : \mathcal{D} \times \mathcal{X} \to \mathcal{Y}$ and a target sample $x_0 \in \mathcal{X}$ with its corresponding ground truth label $y_0 \in \mathcal{Y}$, the attack succeeds and the defense fails when a poisoned training set $D' \in \mathcal{D}$ is found such that $D \oplus D' \leq \tau$ and $f(D', x_0) \neq y_0$. Here $D \oplus D' = |(D \setminus D') \cup (D' \setminus D)|$ is the symmetric distance between two training sets, which is equal to the minimum number of insertions and removals needed to change $D$ to $D'$. $\tau$ is the attack budget denoting the maximum number of poisoned samples allowed.

## 2.2 Proposal: Temporal Modeling of Data Poisoning

In existing threat models for data poisoning, the key characteristic of an attack is the number of poisoned samples: Attacks try to poison (i.e. insert or remove) fewer samples to manipulate model behaviors and defenses try to preserve model utility assuming a bounded number of poisoned samples.

As a result, if attackers can poison more samples than expected with affordable overhead, which is possible in many practical scenarios, they may render existing defenses ineffective in a short time. Towards addressing this issue, we incorporate the concepts of time in modeling data poisoning.

**Definition 2.1** (Timeline). *The timeline $T$ is simply the set of all integers $\mathbb{Z}$, where smaller integers denote earlier points in time and the difference of two integers is associated with the amount of time between them.*

Notably, we use notions of discrete timelines in this paper.

**Definition 2.2** (Data with Birth Dates). *Let $\mathcal{X}$ be the input space, $\mathcal{Y}$ be the label space, every sample is considered to have the form of $(x, y, t)$, where $x \in \mathcal{X}$ is its input, $y \in \mathcal{Y}$ is its label and $t \in T$ is a timestamp, namely its birth date, denoting when the sample becomes available for collection. We use $\mathcal{D}_T = (\mathcal{X} \times \mathcal{Y} \times T)^{\mathbb{N}}$ to denote the space of all possible training sets with timestamps.*

In practice, the birth dates of samples are determined in the data collection process. For example, a company may set the birth dates to the actual creation time of its users' data; When scraping data from the internet, one may use upload dates available on third-party websites; In cases of continuous data collection, the birth dates can also be simply set to the time that samples are first seen.

**Definition 2.3** (Temporal Data Poisoning Attack). *Let $D_{clean} \in \mathcal{D}_T$ be the clean training set, i.e. the training set that would be collected if there were no attacker. The attacker can insert a set of samples $D_{insert} \in \mathcal{D}_T$ to the training set and remove a set of samples $D_{remove} \subseteq D_{clean}$ from the training set, following Assumption 2.4 but with no bound on the numbers of samples inserted or removed. Let $D' = (D_{clean} \setminus D_{remove}) \cup D_{insert}$ be the resulted, poisoned training set and $f : \mathcal{D}_T \times \mathcal{X} \to \mathcal{Y}$ be a learner that maps an input to a label conditioning on a training set with timestamps. Given a target sample $x_0 \in \mathcal{X}$ with a corresponding ground truth label $y_0 \in \mathcal{Y}$, the attack succeeds and the defense fails when $f(D', x_0) \neq y_0$.*

**Assumption 2.4** (Partial Reliability of Birth Dates). *The attacker has **no** capability to **directly** set the birth dates of samples.*

In another word, Assumption 2.4 means that no poisoned sample can have a birth date falling outside the active time of the attacker:

- If an attack starts at time $t_0$, no poisoned sample can have a timestamp smaller than $t_0$ (i.e. $t \geq t_0$ for all $(x, y, t) \in D_{\text{remove}} \cup D_{\text{insert}}$);

- If an attack ends at time $t_1$, no poisoned sample can have a timestamp larger than $t_1$ (i.e. $t \leq t_1$ for all $(x, y, t) \in D_{\text{remove}} \cup D_{\text{insert}}$).

It is worth noting that this does not mean that the attacker has no control over the birth dates of samples. For instance, an attacker may inject poisoned samples at a specific time to control their birth dates or change the birth dates of clean samples by blocking their original uploads (i.e. removing them from the training set) and re-inserting them later.

With the above framework, we now introduce two temporal metrics, namely **earliness** and **duration**, that are used as attack budgets in our temporal modeling of data poisoning. An illustration of both concepts is included in Figure 1.

**Definition 2.5** (Earliness). *Earliness measures how long a poisoning attack started in advance (i.e. how early it is when the attack started). The earliness of an attack is defined as*

$$\tau_{earliness} = t_{end} - \left( \min_{(x,y,t) \in D_{poison}} t \right) + 1,$$

*where $t_{end}$ is the time when the data collection ends, $D_{poison} = D_{remove} \cup D_{insert}$ is the set of poisoned samples, containing all samples that are inserted or removed by the attacker. The earliness of an attack is determined by the poisoned sample with the earliest birth date.*

**Definition 2.6** (Duration). *Duration measures how long a poisoning attack lasted. The duration of an attack is defined as*

$$\tau_{duration} = \left( \max_{(x,y,t) \in D_{poison}} t \right) - \left( \min_{(x,y,t) \in D_{poison}} t \right) + 1,$$

*where $D_{poison} = D_{remove} \cup D_{insert}$ is the set of all poisoned samples. The duration of an attack is determined by the time passed from the earliest birth date of poisoned samples to the latest one.*

These notions for earliness and duration of attacks enable the description of the following new form of robustness that is never available without incorporating temporal concepts.

**Definition 2.7** (Temporal Robustness against Data Poisoning). *Let $D_{clean} \in \mathcal{D}_T$ be the clean training set and $f : \mathcal{D}_T \times \mathcal{X} \to \mathcal{Y}$ be a learner that maps an input to a label conditioning on a training set with timestamps. For a given target sample $x_0 \in \mathcal{X}$ with the ground truth label $y_0$,*

- *$f$ is robust against data poisoning with a maximum earliness of $\tau$ if for all $D_{insert}$ and $D_{remove}$ with $\underline{\tau_{earliness}} \leq \tau$, $f((D_{clean} \setminus D_{remove}) \cup D_{insert}, x_0) = y_0$ .*

- *$f$ is robust against data poisoning with a maximum duration of $\tau$ if for all $D_{insert}$ and $D_{remove}$ with $\underline{\tau_{duration}} \leq \tau$, $f((D_{clean} \setminus D_{remove}) \cup D_{insert}, x_0) = y_0$.*

Informally, a learner or a defense offers temporal robustness against data poisoning if all poisoning attacks with earliness or duration bounded by $\tau$ cannot manipulate its behavior.

Temporal robustness against data poisoning can be more desirable than existing notions, especially for scenarios where attackers may poison more and more samples with affordable overhead: The number of poisoned samples is no longer a good measure of attack efforts, but the proposed concepts of earliness and duration are closely tied to the difficulty of conducting the attack as long as time travels remain fictional.

## 3   A Temporal Robustness Benchmark

In this section, we present a benchmark to support empirical evaluations of temporal robustness by simulating continuous data collection and periodic deployments of updated models.

### 3.1   Dataset

We use News Category Dataset [20, 21] as the base of our benchmark, which contains news headlines from 2012 to 2022 published on HuffPost (https://www.huffpost.com). In News Category Dataset, every headline is associated with a category where it was published and a publication date. Notably, this setting is well aligned with our Assumption 2.4, Partial Reliability of Birth Dates, as a potential attacker may be capable of submitting a poisoned headline but cannot set a publication date that is not its actual date of publication.

Following the recommendation by [20], we use only samples within the period of 2012 to 2017 as HuffPost stopped maintaining an extensive archive of news articles in 2018. To be specific, we preserve only news headlines dated from 2012-02-01 to 2017-12-31 and the resulting dataset contains in total $191939$ headlines and $41$ categories, temporally spanning over $n_{month} = 71$ months. For better intuitions, we include examples of headlines with categories

| Headline | Category | Date |
|---|---|---|
| The Good, the Bad and the Beautiful | WELLNESS | 2012-02-08 |
| Nature Is a Gift | FIFTY | 2015-04-29 |
| How To Overcome Post-Travel Blues | TRAVEL | 2016-06-26 |
| Who Is The Real Shande Far Di Goyim? | POLITICS | 2017-08-27 |

Table 1: Examples of headlines with categories and publication dates from News Category Dataset [20].

and publication dates in Table 1 and the counts of headlines in each month in Figure 2. In addition, while natural distribution shifts over time are not the main focus of this work, it is worth noting that such natural shifts exist evidently, as shown in Figure 3, where we can observe that most categories are only active for a subset of months.

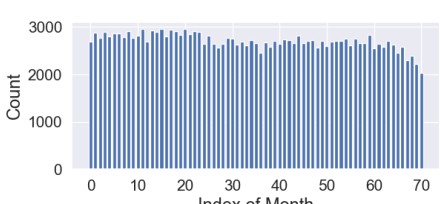

Figure 2: Counts of headlines published in each month. On average, 2703 headlines are published per month.

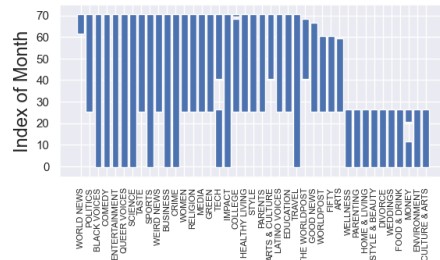

Figure 3: Presences of news categories in different months. Marked positions indicate categories published in the corresponding month.

## 3.2 Evaluation Protocol

Following the notations from Section 2.2, we denote every sample in the form of $(x, y, t)$, where $x \in \mathcal{X}$ is the headline, $y \in \mathcal{Y}$ is its corresponding category and $t \in T$ is its publication date.

We use $D_i$ to denote the set of all samples published in the $i$-th month for $0 \leq i < n_{\text{month}} = 71$.

Let $f : \mathcal{D}_T \times \mathcal{X} \to \mathcal{Y}$ be a learner (i.e. a function that maps a headline to a category conditioning on a training set with timestamps). Recalling the notion of temporal robustness from Definition 2.7, for a given target sample $x_0$ with the ground truth label $y_0$, we use

- $\mathbb{1}_{\tau_{\text{earliness}} \leq \tau} \left[ f, D, (x_0, y_0) \right]$ to denote whether $f$ is robust against data poisoning with a maximum earliness of $\tau$ when $D_{\text{clean}} = D$;

- $\mathbb{1}_{\tau_{\text{duration}} \leq \tau} \left[ f, D, (x_0, y_0) \right]$ to denote whether $f$ is robust against data poisoning with a maximum duration of $\tau$ when $D_{\text{clean}} = D$.

With these notions, we define robust fractions corresponding to bounded earliness and duration of poisoning attacks, which serve as robustness metrics of the temporal robustness benchmark.

**Definition 3.1** (Robust Fraction). *For a given $n_{start}$, which denotes the first month for testing,*

- *the robust fraction of a learner $f$ corresponding to a maximum earliness of $\tau$ is defined as*

$$\frac{\sum_{i=n_{start}}^{n_{month}-1} \sum_{(x,y,t) \in D_i} \mathbb{1}_{\tau_{earliness} \leq \tau} \left[ f, \bigcup_{j=0}^{i-1} D_j, (x,y) \right]}{\left| \bigcup_{j=n_{start}}^{n_{month}-1} D_j \right|};$$

- *the robust fraction of a learner $f$ corresponding to a maximum duration of $\tau$ is defined as*

$$\frac{\sum_{i=n_{start}}^{n_{month}-1} \sum_{(x,y,t) \in D_i} \mathbb{1}_{\tau_{duration} \leq \tau} \left[ f, \bigcup_{j=0}^{i-1} D_j, (x,y) \right]}{\left| \bigcup_{j=n_{start}}^{n_{month}-1} D_j \right|}.$$

Intuitively, here we are simulating a continuous data collection with periodic deployments of updated models, where models can be updated monthly with access to data newly arrived this month and all previous data. In another word, each month of samples will be used as test data individually, and when predicting the categories of samples from the $i$-th month, the model to be used will be the one trained from only samples of previous months (i.e. from the 0-th to the $(i-1)$-th month). The first $n_{\text{start}}$ months are excluded from test data to avoid performance variation due to the scarcity of training samples. Here, with $n_{\text{month}} = 71$ months of data available, we set $n_{\text{start}} = 36$, which is about half of the months. The resulted test set contains a total of 91329 samples, i.e. $\left| \bigcup_{j=n_{\text{start}}}^{n_{\text{month}}-1} D_j \right| = 91329$.

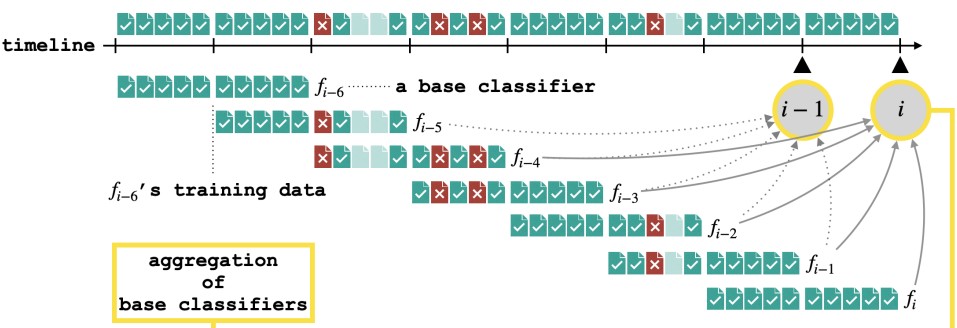

Figure 4: An illustration of temporal aggregation with base coverage $n = 2$ and aggregation size $k = 5$ (i.e. 2 periods of training data per base classifier and 5 base classifiers per aggregation).

## 4 A Baseline Defense with Provable Temporal Robustness

### 4.1 Temporal Aggregation

Here we present temporal aggregation, a very simple baseline defense with provable temporal robustness, deliberately designed for periodic model updates with continuous data collection, to corroborate the usability of our temporal threat model. Considering a process of periodic model updates with continuous data collection, where $i$ is the index of the current period, we use $D_0, D_1, \ldots, D_i$ to denote respectively the sets of samples collected in each of all finished data collection periods. For simplicity, we use $D_{s,t} = \bigcup_{j=s}^{t} D_j$ to denote the union of $D_s$ through $D_t$.

**Definition 4.1** (Temporal Aggregation). *Given a deterministic learner $f$ that maps a training set $D \in \mathcal{D}_T$ to a classifier $f_D : \mathcal{X} \to \mathcal{Y}$ (i.e. a function mapping the input space $\mathcal{X}$ to the label space $\mathcal{Y}$), **temporal aggregation** with base coverage $n$ and aggregation size $k$ computes the prediction for a test sample $x \in \mathcal{X}$ as follows:*

$$\arg\max_{y \in \mathcal{Y}} \sum_{j=i-k+1}^{i} \mathbb{1}\left[f_j(x) = y\right],$$

*where $f_j = f_{D_{j-n+1,j}}$ is the base classifier obtained by applying the deterministic learner $f$ to the union of $D_{j-n+1}, D_{j-n+2}, \ldots, D_j$, $\mathbb{1}[*]$ denotes the indicator function and ties in $\arg\max$ are broken by returning the label with smaller index.*

An illustration of temporal aggregation with base coverage $n = 2$ and aggregation size $k = 5$ is included in Figure 4. In short, for every updating period of temporal aggregation, one trains only one new base model using data collected in the last $n$ periods and uses the majority votes of the latest $k$ base models for predictions. Such efficient updates are favorable in practice.

### 4.2 Provable Temporal Robustness against Poisoning

Intuitively, temporal aggregation offers temporal robustness by limiting the influence of training data within certain intervals of time to the aggregations at inference time. Now we present theoretical results showing its provable temporal robustness. For simplicity, we use 'period' as the unit of time in this section and use $TA([D_0, \ldots, D_i], x)$ to denote the prediction of temporal aggregation on the input $x \in \mathcal{X}$, assuming $D_0, \ldots, D_i$ are sets of samples collected in previous periods.

**Theorem 4.2** (Provable Temporal Robustness against Data Poisoning with Bounded Earliness). *Let $D_0, D_1, \ldots, D_i$ be sets of samples collected in different periods, where $i$ is the index of the current period. Let $n$ and $k$ be the base coverage and aggregation size of temporal aggregation. For any $x \in \mathcal{X}$, let $y = TA([D_0, \ldots, D_i], x)$. For any $\tau \in \mathbb{N}$, if for all $y' \in \mathcal{Y} \setminus \{y\}$,*

$$\sum_{j=i-k+1}^{i-\tau} \mathbb{1}\left[f_j(x) = y\right] \geq \sum_{j=i-k+1}^{i-\tau} \mathbb{1}\left[f_j(x) = y'\right] + \min(\tau, k) + \mathbb{1}[y > y'], \tag{1}$$

then we have $TA([D'_0, \ldots, D'_i], x) = y$ for any $D'_0, \ldots, D'_i$ reachable from $D_0, \ldots, D_i$ after some data poisoning attack with a maximum earliness of $\tau$.

*Proof.* The idea of this proof is straightforward. Given a maximum earliness of $\tau$, only data from the last $\tau$ periods may be poisoned and thus only the last $\tau$ base classifiers may behave differently after such an attack. With this observation, it is clear that the left-hand side of Inequality 1, $\sum_{j=i-k+1}^{i-\tau} \mathbb{1}[f_j(x) = y]$ is a lower bound for the number of base classifiers in the aggregation that predicts label $y$. Similarly, $\sum_{j=i-k+1}^{i-\tau} \mathbb{1}[f_j(x) = y'] + \min(\tau, k)$ is an upper bound for the number of base classifiers in the aggregation that predicts label $y'$, since no more than $\min(\tau, k)$ base classifiers in the aggregation can be affected. The remaining term in Inequality 1, $\mathbb{1}[y > y']$ is due to the tie-breaking rule of temporal aggregation. Thus when Inequality 1 holds for $y'$, we know $TA([D'_0, \ldots, D'_i], x) \neq y'$. Since Inequality 1 holds for all $y' \neq y$, we have $TA([D'_0, \ldots, D'_i], x) = y$ and the proof is completed. $\square$

**Theorem 4.3** (Provable Temporal Robustness against Data Poisoning with Bounded Duration). *Let $D_0, D_1, \ldots, D_i$ be sets of samples collected in different periods, where $i$ is the index of the current period. Let $n$ and $k$ be the base coverage and aggregation size of temporal aggregation. For any $x \in \mathcal{X}$, let $y = TA([D_0, \ldots, D_i], x)$. For any $\tau \in \mathbb{N}$, if for all $y' \in \mathcal{Y} \setminus \{y\}$ and all $s \in \mathbb{N}$ with $i - k + 1 \leq s \leq i$,*

$$\sum_{j=i-k+1}^{s-1} \mathbb{1}[f_j(x) = y] + \sum_{j=s+\tau+n-1}^{i} \mathbb{1}[f_j(x) = y]$$

$$\geq \sum_{j=i-k+1}^{s-1} \mathbb{1}[f_j(x) = y'] + \sum_{j=s+\tau+n-1}^{i} \mathbb{1}[f_j(x) = y'] + \min(\tau + n - 1, k) + \mathbb{1}[y > y'] \quad (2)$$

*then we have $TA([D'_0, \ldots, D'_i], x) = y$ for any $D'_0, \ldots, D'_i$ reachable from $D_0, \ldots, D_i$ after some data poisoning attack with a maximum duration of $\tau$.*

*Proof.* With an upper bound of $\tau$ for the duration, an attack cannot affect simultaneously any two periods that are at least $\tau$ periods away from each other, and therefore no two base classifiers that are at least $\tau + n - 1$ away from each other can be both poisoned. Thus, if assuming $f_s$ is the first base classifier affected by the attack, $f_j$ with $i - k + 1 \leq j \leq s - 1$ or $j \geq s + \tau + n - 1$ must be free from any impact of the attack, and therefore the left-hand side of Inequality 2 is a natural lower bound for the number of base classifiers in the aggregation that predicts label $y$. In addition, since no two base classifiers that are at least $\tau + n - 1$ away from each other can be both poisoned, $\sum_{j=i-k+1}^{s-1} \mathbb{1}[f_j(x) = y'] + \sum_{j=s+\tau+n-1}^{i} \mathbb{1}[f_j(x) = y'] + \min(\tau + n - 1, k)$ on the right-hand side of Inequality 2 becomes an upper bound for the number of base classifiers in the aggregation that predicts label $y'$. Again, $\mathbb{1}[y > y']$ is due to the tie-breaking rule of temporal aggregation. Consequently, we know that Inequality 2 is true for a certain $y'$ and a certain $s$ suggests $TA([D'_0, \ldots, D'_i], x) \neq y'$, assuming the first base classifier affected is $f_s$. Since it it true for all $y' \neq y$ and all $s$, we have $TA([D'_0, \ldots, D'_i], x) = y$ and the proof is completed. $\square$

**Implications and applications of Theorem 4.2 and 4.3:** The key conditions used in these two theorems, Inequality 1 and 2, are both computationally easy given predictions from base classifiers: For any sample at inference time, by simply examining the predictions from base classifiers, one can obtain lower bounds for both earliness and duration of data poisoning attacks that stand a chance of changing the prediction from temporal aggregation.

In addition, it is worth noting that Theorem 4.2 and 4.3 do not assume whether $D_0, \ldots, D_i$ are all clean or potentially poisoned. Given that in the temporal threat model of data poisoning, $D'_0, \ldots, D'_i$ is reachable from $D_0, \ldots, D_i$ after some poisoning attack with a maximum earliness (or duration) of $\tau$ **if and only if** $D_0, \ldots, D_i$ is also reachable from $D'_0, \ldots, D'_i$ after some poisoning attack with a maximum earliness (or duration) of $\tau$, these theorems can be applied regardless of whether the training sets are considered clean or potentially poisoned: With clean training data, we obtain lower bounds of earliness and duration required for a hypothetical poisoning attack to be effective; With potentially poisoned data, we get lower bounds of earliness and duration required for a past poisoning attack to be effective. In this work, we apply them for evaluation of temporal aggregation on the temporal robustness benchmark from Section 3, where data are assumed poison-free.

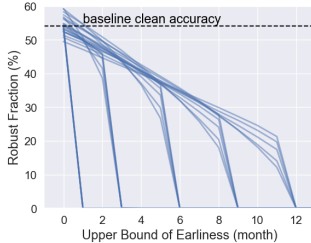
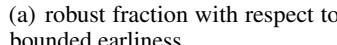
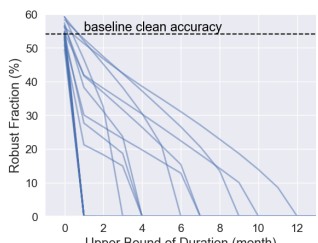

(a) robust fraction with respect to bounded earliness

(b) robust fraction with respect to bounded duration

Figure 5: An overview of the performance and temporal robustness of temporal aggregation with with base coverage $n \in \{1, 3, 6, 9, 12\}$ and aggregation size $k \in \{6, 12, 18, 24\}$, a total of 20 combinations of hyperparameters. Detailed views of individual runs can be found in Figure 6 and Figure 7.

# 5 Evaluation of the Baseline Defense

## 5.1 Evaluation Setup

Given that the temporal robustness benchmark simulates a setting with monthly updated models, we consider each month a separate period for temporal aggregation (i.e. one base classifier per month) and use 'month' as the unit of time. For the base learner (i.e. the learning of each base classifier), we use the pre-trained RoBERTa [17] with a BERT-base architecture as a feature extractor, mapping each headline into a vector with 768 dimensions. We optimize a linear classification head over normalized RoBERTa features with AdamW optimizer [18] for 50 epochs, using a learning rate of 1e-3 and a batch size of 256. To make the base learner deterministic, we set the random seeds explicitly and disable all nondeterministic features from PyTorch [24].

## 5.2 Establishing Baseline Accuracy

To establish baselines for assessing the performance and temporal robustness of temporal aggregation, we present in Table 2 clean accuracy, i.e. robust fraction from Definition 3.1 with $\tau = 0$, of the base learner trained on varying months of latest data. Corroborating the effects of natural distribution shifts, among settings evaluated, the average clean accuracy over all categories reaches its peak when using the latest one month of data for training, thus we use the corresponding value,

| Category | $n = 1$ | $n = 3$ | $n = 6$ | $n = 9$ | $n = 12$ |
|---|---|---|---|---|---|
| **all** | 54.09% | 53.77% | 53.47% | 53.43% | 53.15% |
| WORLD NEWS | 59.00% | 46.59% | 36.79% | 31.39% | 25.56% |
| POLITICS | 62.13% | 57.52% | 55.36% | 54.31% | 53.96% |
| TRAVEL | 52.30% | 56.65% | 59.46% | 61.63% | 62.26% |
| COLLEGE | 42.30% | 49.30% | 51.68% | 54.20% | 57.28% |

Table 2: Clean accuracy of the base learner trained on the latest $n$ months of samples.

$54.09\%$ as the baseline clean accuracy in our evaluation. In addition, here we observe that different categories can have different degrees of temporal distribution shifts and different data availability: For some categories including WORLD NEWS/POLITICS, training on earlier data reduces accuracy, indicating effects of temporal distribution shifts; For some categories including TRAVEL/COLLEGE, training on earlier data improves accuracy, indicating the effects from using more data.

## 5.3 Performance and Robustness

In Figure 5, we include an overview of the performance and robustness of temporal aggregation with base coverage $n \in \{1, 3, 6, 9, 12\}$ and aggregation size $k \in \{6, 12, 18, 24\}$, a total of 20 combinations of hyperparameters. For clarity, we defer the study of hyperparameters to Section 5.4.

From Figure 5, we observe that temporal aggregation is capable of providing decent temporal robustness with minimal loss of clean accuracy. For example, with $n = 1$ and $k = 24$, it is provably robust for a sizable share (with respect to the baseline clean accuracy) of test samples against poisoning attacks that started no more than $8 \sim 11$ months in advance (earliness) and attacks that lasted for no more than $6 \sim 9$ months (duration).

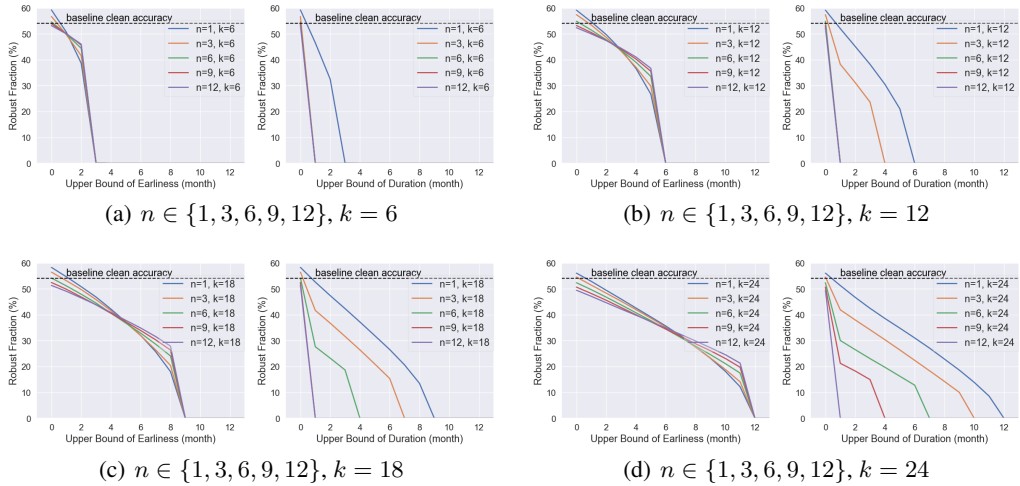

Figure 6: The effects of base coverage $n$ to the performance and temporal robustness.

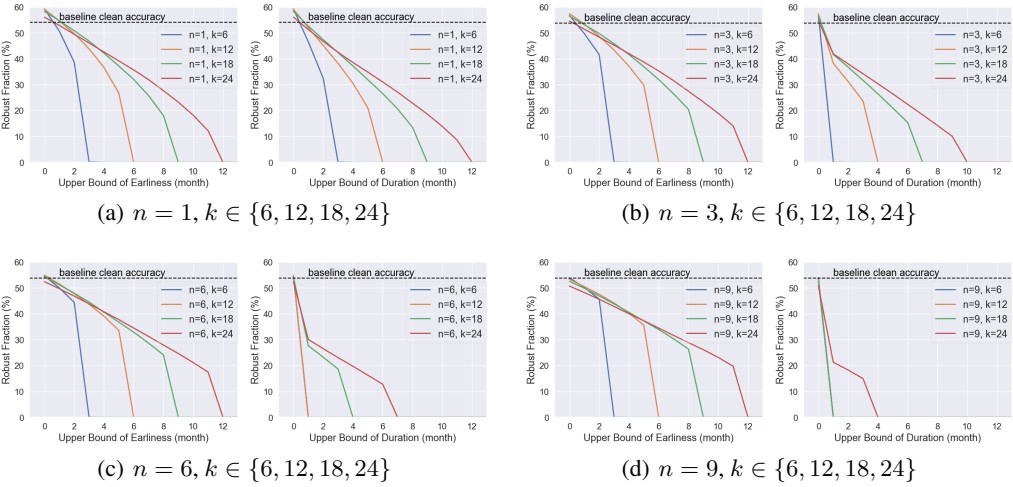

Figure 7: The effects of aggregation size $k$ to the performance and temporal robustness.

## 5.4 Hyperparameters for Temporal Aggregation

**The effects of base coverage $n$:** In Figure 6, we compare the robust fraction of temporal aggregation with different base coverage $n$. Intuitively, the choice of $n$ involves a trade-off between training set size and distribution shifts: A larger $n$ increases training set size, which is positive to clean accuracy, but also enlarges distribution shifts, as more samples from early-time will be included, which is negative to clean accuracy. In this case, however, we observe a mild but negative impact from increasing $n$ to performance, consistent with previous observations from Table 2, again indicating the presence of strong natural distribution shifts over time. Regarding temporal robustness, Figure 6 suggests that a larger $n$ significantly reduces temporal robustness of temporal aggregation against poisoning attacks with bounded duration, which is expected as an attacker can affect more base classifiers within the same duration.

**The effects of aggregation size $k$:** In Figure 7, we present a comparison of temporal aggregation with different aggregation size $k$, where we observe that a larger $k$ has a rather mild negative impact on performance but greatly facilitates temporal robustness with both bounded earliness and duration. This is because, with a larger $k$, more base classifiers will be utilized in the aggregation. On the one hand, these extra base classifiers are trained on earlier data with larger natural distribution shifts from

test distribution, hence the reduced clean accuracy; On the other hand, extra base classifiers dilute the impact from poisoned base classifiers to the aggregation, hence the improved robustness.

# 6 Related Work

The proposed defense, temporal aggregation, is inspired by aggregation-based certified defenses for the traditional modeling of data poisoning, with a shared spirit of using voting for limiting the impacts of poisoned samples. [16, 39, 40, 26, 38] assume deterministic base learners and guarantee the stability of predictions with bounded numbers of poisoned samples. [14, 6] provides probabilistic guarantees of the same form with small, bounded error rates using potentially stochastic learners.

# 7 Discussion

**Temporal Aggregation introduces little-to-none extra training cost in many practical cases.** In reality with continuous data collection, a common practice is to re-train/update models using new data every once in a while because model performance degrades as time evolves. As mentioned in Section 4.1, to use Temporal Aggregation, one still only need to train one classifier per updating period since previous base classifiers can be reused, which incurs minimal/no extra training overhead.

**The increase of inference overhead can be small depending on the design of base classifiers.** Naturally, using more base classifiers (i.e. using a larger $k$) increases the inference overhead. However, the extra inference overhead can be small depending on the designs of base classifiers. An example design is to fine-tune only the last few layers of the model instead of training entirely from scratch. If one uses a shared, pre-trained feature extractor and only trains a task head for each base classifier (as in our Section 5), the shared feature extractor only needs to be forwarded once for inference and the extra overhead can be minimized.

**Using both time and the number of poisoned samples to characterize attack budgets.** While this work focuses on the benefits of incorporating temporal concepts, it is worth noting that earliness and duration can be combined with the number of poisoned samples in bounding attack budgets. As an example, our baseline defense Temporal Aggregation is not only a provable defense against attacks with bounded earliness and duration but also a provable defense with respect to a bounded number of poisoned samples (when using a small base coverage $n$). This follows directly from previous aggregation defenses [16, 39] targeting attacks with the number of poisoned samples bounded.

**Potential opportunities for other defenses.** The temporal threat model of data poisoning may offer new angles to improve the usability of existing poisoning defenses for different stages. For detection-based defenses [33, 9, 25, 34, 37, 23, 44] targeting the early stages where the models have not been trained, the temporal threat model may delay the success of poisoning attacks and allow detection-based defenses to operate on more data. Another promising direction is to locate poisoned samples after the attack succeeds [31], which targets later stages where the models have been deployed. If one can reverse poisoning attacks shortly after they succeed, incorporating the temporal threat model can help to prolong every attack process and may eventually ensure a low exploitability of poisoning attacks.

# 8 Conclusion

Incorporating timestamps of samples that are often available but neglected in the past, we provide a novel, temporal modeling of data poisoning. With our temporal modeling, we derive notions of temporal robustness that offer meaningful protections even with unbounded amounts of poisoned samples, which is never possible with traditional modelings. We design a temporal robustness benchmark to support empirical evaluations of the proposed robustness notions. We propose temporal aggregation, an effective baseline defense with provable temporal robustness. Evaluation verifies its temporal robustness against data poisoning and highlights the potential of the temporal threat model.

## Acknowledgements

This project was supported in part by Meta grant 23010098, NSF CAREER AWARD 1942230, HR001119S0026 (GARD), ONR YIP award N00014-22-1-2271, Army Grant No. W911NF2120076, NIST 60NANB20D134, the NSF award CCF2212458, a capital one grant and an Amazon Research Award.

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
