# OpenReview forum: "Temporal Robustness against Data poisoning"
_NeurIPS.cc/2023/Conference — NeurIPS 2023 poster_

### Official Review · Reviewer_exCr · 2023-07-04

**Soundness:** 3 good
**Presentation:** 3 good
**Contribution:** 2 fair
**Rating:** 5
**Confidence:** 5

**Summary:**

This paper considers provable defenses in a threat model where attackers can inject arbitrarily many poisoning points. Traditional defenses that rely on the assumption of dealing with a low fraction of poisoning points can then be easily broken in the new threat model. Regarding this issue, the authors propose to leverage the birth date of the data collected and defined two temporal metrics of earliness and duration of an attack. Then, for attacks with bounded earliness and duration, the proposed temporal framework can provide a provable certificate on predictions, even when the attackers can inject arbitrarily many poisoning samples. The authors empirically validated the performance of the defense by establishing a baseline on the news category dataset.

**Strengths:**

1. The presentation of the paper is clear.
2. The proposed claims are well-supported through both theoretical and empirical validations.

**Weaknesses:**

1. It is unclear how realistic the considered threat model is in practice.
2. The considered threat model allows the existence of strong attackers that continuously inject poisoning points with unbounded earlierness and duration and make the provable defenses completely fail.

**Questions:**

This paper is easy to read and raises an interesting research question on poisoning defense in a new dimension. However, I have a major concern about the threat model.
1. Will the considered threat model be realistic in practice? The authors claimed that it is possible for attackers to post more poisoning points online, but this is based on the assumption that the generation of individual poisoning points is very cheap and can be automated at a low cost. I could not think of a practical application scenario where this is likely to be true. In fact, all poisoning points need to be modified deliberately to be effective with some decent optimization strategies, and so this process can be costly. In addition, the generated poisoning points may also need to be coordinated, making the task harder for full automation. Recent work shows how poisoning can be practical [1], but the key assumption is these attacks are feasible to manipulate a low fraction of poisoning points and controlling exceedingly many numbers of poisoning points would be practically infeasible.

2. The considered threat model allows any cautious attacker to start the attack early as possible and continue to poison the dataset as long as possible. The provable defense can be helpful for the initial period of time, as there can be some leverage from datasets that are sampled prior to the start of the attack. However, this leverage will be completely lost later. So, it is unclear to me how significant the result can be, as it means in the end, there are exceedingly many poisoning points available and any viable defenses can be broken.

Tow minor questions are:
1. what is the main motivation behind making the base learner deterministic in the experiments?
2. the authors mention that the birth date is partially reliable. I am wondering if the fact that attackers can block the collection of clean samples (and hence their birth dates) will increase the effectiveness of the attack (e.g., making the attack start early in a passive manner). My current understanding is whatever strategy is used (e.g., injection of new points or blocking and inserting clean points later) are all considered as the start of the attack, and therefore, the robustness guarantee will be the same.

[1] Carlini et al., "Poisoning Web-Scale Training Datasets is Practical", ArXiv.

--------------------------------------

This is to acknowledge that I have read the authors rebuttals carefully and my concerns are addressed.

---

> ### Author Rebuttal · Authors · 2023-08-10
>
> Thank you for recognizing our paper as clear and well-supported! We value your comments a lot in improving the presentation of our paper.
>
> **1. ‘The authors claimed that it is possible for attackers to post more poisoning points online, but this is based on the assumption that the generation of individual poisoning points is very cheap and can be automated at a low cost’**
>
> We understand your confusion and we will revise the expressions to better reflect our positions. To clarify, by ‘attackers in many practical scenarios can poison more and more samples with fairly affordable overhead’ and the following examples, what we want to emphasize is that they can poison more samples than **what existing defenses can tolerate**, rather than that they can indeed poison astronomical amounts of samples (i.e. what we wanted to mean is that the poisoned samples are exceedingly many ONLY when being compared with the numbers we can defend against so far).
>
> Notably, the fractions/numbers of poisoned samples defenses may tolerate can be quite limited (supported empirically by [1, 2, 3] and theoretically by [4], just for examples).
>
> We see how our existing descriptions can be confusing, and we will incorporate your comments for revisions.
>
> [1] Carlini et.al., Poisoning Web-Scale Training Datasets is Practical.
>
> [2] Souri et.al., Sleeper Agent: Scalable Hidden Trigger Backdoors for Neural Networks Trained from Scratch.
>
> [3] Geiping et.al., Witches' Brew: Industrial Scale Data Poisoning via Gradient Matching.
>
> [4] Wang et.al., Lethal Dose Conjecture on Data Poisoning.
>
> **2. 'The considered threat model allows the existence of strong attackers that continuously inject poisoning points with unbounded earlierness and duration and make the provable defenses completely fail'**
>
> Since the key of our threat model is to use earliness and/or duration to measure attack budgets, it is natural that poisoning attacks with unbounded budgets (essentially no limit at all) can break the provable defenses. However, we believe this, *by itself*, does not constitute a weakness of our threat model since similar arguments apply to essentially any threat model (e.g. there will also be infinitely broken points for previous data poisoning threat models if one has no constraint on the number of poisoned samples).
>
> We suggest a different angle to inspect the usefulness of our threat model v.s. existing ones, by considering what different metrics (earliness, duration, and number of poisoned samples) mean from the eyes of an attacker.
>
> First, we consider the number of poisoned samples, as in previous threat models for data poisoning. As we previously mentioned, the number of poisoned samples existing defenses can tolerate can be quite limited (supported empirically by [1, 2, 3] and theoretically by [4]). Admittedly, there can be some cases where the attacker simply cannot meet the requirement, but it is unlikely unless it is a very simple task (e.g. MNIST) or the victim has (or has the clean data to train) a strong pre-trained model available that is assumed clean. In other cases where the attacker can inject the required amounts of poisoned samples, the attack will succeed as soon as the victim’s next round of model updates.
>
> The role of earliness is to delay the success of the attack when the number of poisoned samples cannot stop the attack entirely. In our evaluations, the earliness requirement for successful attacks varies from a couple of months to almost a year, which we hope can be further increased with better defenses. If I were the attacker, the need to wait for many months (and maybe years depending on tasks and future improvements on defenses) can not only be emotionally painful but also prohibits attack goals that are sensitive to timing (e.g. sabotage new models from competitors to promote one’s own) and increases the difficulty of adjusting attack methods based on attack results. Duration, on the other hand, enforces attackers to be active for a longer period of time, which not only increases their workload but also potentially makes them leave more traces so that they have to cost more to keep the attacks unnoticed.
>
> In addition, earliness and duration can be used jointly with the number of poisoned samples to raise the cost/difficulty of successful poisoning attacks. As an example, our baseline defense, while being a provable defense against attacks with bounded earliness and duration, is also a provable defense with respect to a bounded number of poisoned samples when using small n. This follows directly from previous aggregation defenses like DPA [5] for existing poisoning threat models.
>
> [5] Alexander Levine and Soheil Feizi. Deep partition aggregation: Provable defenses against general poisoning attacks. ICLR 2021
>
> **3. ‘what is the main motivation behind making the base learner deterministic in the experiments?’**
>
> As discussed in Section 6 of the paper, the baseline defense is inspired by aggregation-based defenses for the traditional threat model of data poisoning (i.e. using the number of poisoned samples to measure attack budgets). We follow [13, 23, 33, 34] cited in the paper, which are all defenses that use deterministic base learners. By making the base learner deterministic, we can make the entire algorithm deterministic and the computed certified robustness can be deterministic, which improves overall efficiency by not requiring Monte Carlo sampling to estimate the mean of output distributions.
>
> **4. ‘My current understanding is whatever strategy is used (e.g., injection of new points or blocking and inserting clean points later) are all considered as the start of the attack, and therefore, the robustness guarantee will be the same’**
>
> Yes, we believe your understanding here is accurate. Injecting a new sample and removing (blocking) an existing sample are both considered the start of the attack.
>
>
> **We look forward to your follow-up comments.**
>
> **Thank you for helping us improve our paper.**

---

> > ### Comment · Reviewer_exCr · 2023-08-13
> > **Thanks for the rebuttal**
> >
> > Thank you very much for the rebuttal. After reading your response, my initial major concern has been addressed by taking a different angle to look at this paper. I have some follow-up comments/questions before reevaluating my score.
> >
> > > Poisoning a greater number of samples than what is tolerated by existing learners and defenses.
> >
> > I believe revising the statement (for instance, in the second paragraph of the introduction) to explicitly convey that the authors refer to the attackers' capability to generate (slightly) more samples than what is tolerable by current learners and defenses, rather than an arbitrarily high number of poisoning samples, will help the audience form a more accurate expectation regarding the contribution made in this paper. From this perspective, the paper indeed introduces a new temporal dimension, providing defenders with additional means to leverage.
> >
> > Could the authors provide more precision on how existing works support the claim that attackers can easily poison more samples than what is tolerable by current defenses/learners? With reference to the provided source [1], one aspect that might apply here is that attacks often require only 0.001% of poisoning points to achieve certain attack goals, whereas manipulating $\geq$ 0.01% of poisoning points can be accomplished with 10K USD. Also, I am not entirely clear on how the result of the lethal dose conjecture is applied here. My understanding of this theoretical result is that it highlights the inherent vulnerability of a test sample to poisoning, determined by the optimal sample complexity. Are the authors approaching this from the perspective that, for certain test samples, their sample complexities can be quite large, thereby allowing attackers to easily manipulate more samples than their inherently tolerable number (inversely proportional to the optimal sample complexity)? I believe these clarifications are important and I also encourage the authors to incorporate these details into the paper, as they are crucial for providing a clear motivation for the entire paper.
> >
> > > The considered threat model fails the proposed defense after some period of time.
> >
> > With the aforementioned concern addressed, this issue is somewhat alleviated. I am imagining two scenarios where the proposed approach proves useful, primarily by providing an additional layer of protection. The author's response offers insights into how this additional layer of protection comes into effect. Firstly, the current defenses are not flawless, and the tolerable amount of poisoning samples has not yet reached the limits that are suggested by the lethal dose conjecture. Moreover, attackers can manipulate only below this limit due to resource constraints. In this scenario, it is conceivable to design improved defenses such that models in each time interval are robust to poisoning, and the benefits of the proposed work lie in buying some extra time to develop more effective defenses. The second type of scenario might align with the authors' initial thoughts, where attackers can consistently manipulate slightly or even significantly more samples than what is tolerable within the limits. In this case, there might be no hope for designing long-term defenses, but the proposed approach does provide some time to propose alternative non-technical countermeasures, such as legislative actions. By the way, could the authors comment on what could be done for the second scenario, where attackers have already bypassed any temporal defenses (e.g., suggest some possible countermeasures)?
> >
> > Lastly, I encourage the authors to provide discussions (including the examples provided in the response) on this matter in the revised version.

---

> > > ### Author Response · Authors · 2023-08-14
> > > **Thanks for taking your time to respond our rebuttal!**
> > >
> > > Thank you very much for responding to our rebuttal!
> > >
> > > Some of your angles are very interesting and we will make sure to use them to improve our paper.
> > >
> > > >I believe revising the statement (for instance, in the second paragraph of the introduction) to explicitly convey that the authors refer to the attackers' capability to generate (slightly) more samples than what is tolerable by current learners and defenses, rather than an arbitrarily high number of poisoning samples, will help the audience form a more accurate expectation regarding the contribution made in this paper.
> > >
> > > We agree with you and we will revise every one of the relevant statements to make it crystal clear, as we promised in the rebuttal. Thanks again for pointing it out, and sorry for the confusion since we should have presented the statements much more accurately.
> > >
> > > >Could the authors provide more precision on how existing works support the claim that attackers can easily poison more samples than what is tolerable by current defenses/learners?
> > >
> > > We are glad to do so since we didn't get to include it in the rebuttal due to character limits.
> > >
> > > [1] supports the claim primarily by providing practical poisoning approaches for the case of *web-scale training datasets**, showing that it is practical/inexpensive to poison more than 0.01% of many recently published large datasets. Given that common use cases of these datasets is unsupervised/semi-supervised (pre-)training, they refer to the results from [5, 6]: [5] showcases successful attacks to semi-supervised learning with 0.1% of the dataset being poisoned and [6] showcases successful attacks to contrastive methods with 0.01% (for backdoor attacks) and 0.0001% (for targeted poisoning attacks) of data being poisoned. Comparing these two parts jointly support the claim that attackers can at least in some practical cases easily poison more samples than what is tolerable by current defenses/learners. (similar to the understanding of the reviewer)
> > >
> > >
> > > [2] and [3] support the claim that the fractions of samples defenses may tolerate can be quite limited by showcasing successful backdoor and poisoning attacks with no more than 1% (on CIFAR-10) and 0.05% (on ImageNet) of poisoned data, with the backdoor attacks from [2] evaluated against 6 representative defenses at the time and the poisoning attacks from [3] evaluated against filtering-based defenses and differential privacy (as a defense).
> > >
> > >
> > > For [4], i.e. lethal dose conjecture, it also supports that the fractions of samples defenses may tolerate can be quite limited, but from a theoretical angle. The conjecture suggests that for a given test sample, the maximum number of poisoned samples any defense can tolerate (e.g. predicting correctly with a certain probability on this test sample) can be inversely proportional to the sample complexity (i.e. the number of clean samples needed to predict this test sample). Thus even with a sample complexity (e.g. > 1000) that is not considered large in many cases, already one can only tolerate a small fraction (e.g. O(1/1000) ~ 0.1%) of poisoned samples if the conjecture holds in this case.
> > >
> > >
> > > We will formalize these discussions when including them in the revision, and we would like to hear if you have any suggestions.
> > >
> > >
> > > [1] Carlini et.al., Poisoning Web-Scale Training Datasets is Practical.
> > >
> > > [2] Souri et.al., Sleeper Agent: Scalable Hidden Trigger Backdoors for Neural Networks Trained from Scratch.
> > >
> > > [3] Geiping et.al., Witches' Brew: Industrial Scale Data Poisoning via Gradient Matching.
> > >
> > > [4] Wang et.al., Lethal Dose Conjecture on Data Poisoning.
> > >
> > > [5] Carlini. Poisoning the unlabeled dataset of {Semi-Supervised} learning.
> > >
> > > [6] Carlini et.al. Poisoning and backdooring contrastive learning.
> > >
> > > >By the way, could the authors comment on what could be done for the second scenario, where attackers have already bypassed any temporal defenses (e.g., suggest some possible countermeasures)?
> > >
> > > Thank you for sharing your thoughts on this! We find the first part of your insights interesting and it was not on our radar. We will include this in the paper.
> > >
> > >
> > > For the second one, I believe it can be a new angle for advancing poisoning defenses, with potential applications/improvements of both old and new techniques. An example of the (relatively) old techniques will be detection-based defenses [28, 6, 22, 29, 32, 20, 38 cited in the paper], as there may be more to exploit when poisoned samples cover a fair share of the timeline. A relatively new angle is [7] that locates poisoned samples after the attack succeeds, which may deactivate the poison shortly after it succeeds. With improvements, this can be a very powerful tool within our temporal threat models.
> > >
> > > [7] Shan et al. Poison forensics: Traceback of data poisoning attacks in neural network.
> > >
> > >
> > > **Thank you again for these insights! We will incorporate all of these into our paper and we welcome any of your follow-ups!**

---

> > > > ### Author Response · Authors · 2023-08-21
> > > >
> > > > Dear Reviewer exCr,
> > > >
> > > > Thank you again for reviewing and responding to us.
> > > > Since today is the last day of author-reviewer discussion, we would kindly ask you, if possible, to let us know if you have any comments given our previous responses.
> > > >
> > > > Thank you for your time and insights.

---

> > > > > ### Comment · Reviewer_exCr · 2023-08-21
> > > > > **Thanks for the reminder**
> > > > >
> > > > > Thank you for the reminder, and I apologize for not responding in a timely manner. I have reviewed your rebuttal, and the response makes sense to me. I have no further concerns. While I have adjusted my score, I haven't raised it to a higher level as I am still confirming the significance of this work. During the discussion phase, I will carefully gather input from other reviewers to ensure a well-informed final rating. My gratitude to the authors for providing a detailed rebuttal.

---

> > > > > > ### Author Response · Authors · 2023-08-21
> > > > > >
> > > > > > Thank you for responding! Good to know that we have addressed all your concerns!
> > > > > >
> > > > > > We understand if you do not want to make a rush decision about the final score and you do not have to.
> > > > > >
> > > > > > Obviously as authors we do believe that the new threat model we propose are important because it highlights new dimensions that existing and future defenses can leverage to limit the powers of attackers. While it is new, we already see that it is promising from our baseline defenses, from potential new use cases of detection-based defenses, and from the potential combos with [7] that locates poisoned samples after the attack succeeds. Since this is the very beginning, we also expect the impact of this threat model to increase over time with more studies that follow.
> > > > > >
> > > > > > However we admit that the judgement regarding future impact of a work can usually be subjective and we respect the reviewer's judgement. In the end, we just want to thank you again for your feedbacks. Have a nice day!
> > > > > >
> > > > > > [7] Shan et al. Poison forensics: Traceback of data poisoning attacks in neural network.

---

> > > > > > > ### Comment · Reviewer_exCr · 2023-08-21
> > > > > > >
> > > > > > > Thanks for the last-minute response. These additional clarifications will also help other reviewers judge the significance of this work, which I hope will lead to a well-informed decision after the discussion stage. Thank you!

---

### Official Review · Reviewer_EzrD · 2023-07-05

**Soundness:** 2 fair
**Presentation:** 2 fair
**Contribution:** 3 good
**Rating:** 6
**Confidence:** 5

**Summary:**

This paper considers temporal information for data poisoning attacks, i.e., when an attack is crafted, and how long it lasts. The authors further propose a temporal aggregation defense (similar to data aggregation defenses), which relies on training a number of base classifiers (on data within a certain period of time), such that the aggregation of clean classifiers would outweigh that of poisoned classifiers to assure robustness of a specific test sample.

**Strengths:**

- the idea of considering birthdates and duration of data samples is interesting;
- Figure 1 and Figure 4 clearly illustrate the main idea of this paper;
- The temporal aggregation algorithm is a simple but intriguing idea.




**Weaknesses:**

**Practicality of the temporal aggregation defense**:
- It seems that training many base classifiers is quite expensive, especially on large models and datasets. I am wondering if there is a trade-off between extra time costs and certified robustness with respect to the choice of $n$ and $k$?
- Similar to the data aggregation algorithm, I assume that the temporal aggregation algorithm might also reduce the final test accuracy. However, in the experiments, the authors do not show "clean" accuracy (i.e., normal training on the entire dataset) and I do not see a clear comparison or tradeoff.
- Following the above question, Table 2 indeed shows an accuracy $\approx 50$%. Is this the normal result you would get by training on this dataset? In this case, the results on a weak classifier seem not very convincing to me.

**Experiments do not verify all the claims**:
- The authors claim that "temporal aggregation provides protection against unbounded amounts of poisoned samples". This seems not true as the defense would only succeed (or provide certified robustness) if the number of clean base classifiers outweighs that of poisoned classifiers. If there are infinite poisoned samples (thus your training set contains poisoned data only in the extreme case), we can easily verify that temporal aggregation would not provide any protection.
- In Figures 5 and 6, it seems that the proposed algorithm can only certify 60% of the test data at most, which is also contradictory to the "unbounded" claim.
- From lines 275-278, the authors say "it Is provably robust...", what do you mean by provably robust in this case? Does a 60% robust fraction count as provably robust?
- Finally, the authors claim to study data poisoning attacks but do not examine the defense against any real attacks.

**Questions:**

Please see my questions in the above section.

**Limitations:**

Limitations are not addressed.

---

> ### Author Rebuttal · Authors · 2023-08-10
>
> Thank you! Glad to know that the ideas about birth dates of data are found interesting and our baseline defense is considered simple and intriguing. We also appreciate your questions as they indicate where we can improve our presentations. Below are our answers.
>
> **1. Practicality of the temporal aggregation defense**
>
> **1.1 ‘trade-off between extra time costs and certified robustness to n and k?’**
>
> The message is three-fold:
> >(1) The baseline defense introduces little-to-none extra training cost in many practical cases;
> >
> >(2) Overall, using more base classifiers tends to improve certified robustness while increasing inference overhead;
> >
> >(3) The increase of inference overhead can be small depending on the design of base classifiers (e.g. a shared pre-trained feature extractor + different task heads for different base classifiers).
>
> (1) For training cost: In reality with continuous data collection, a common practice is to re-train/update models using new data every once in a while because the old models’ performance degrades as time evolves. As we mentioned in lines 190-192 of the paper, one only needs to train one new base classifier per updating period since most of the previous base classifiers can be reused, which incurs minimal/no extra training overhead.
>
> (2) As discussed in Section 5.4, the choice of n involves a trade-off between training set size and distribution shifts (supported by Figure 6) and a larger k tends to improve certified robustness (supported by Figure 7). Meanwhile, a larger k indeed increases the inference overhead as we use more base classifiers.
>
> (3) However, the extra inference overhead can be small depending on the designs. For instance, another common practice is to fine-tune only the last few layers of the model instead of re-training entirely from scratch. If one uses a shared, pre-trained feature extractor and only trains a task head for each base classifier (as we do in our experiments), the shared feature extractor only needs to be forwarded once for inference, minimizing the extra overhead.
>
> While the baseline defense is already practical to some extent, we agree that it is not yet ideal and look forward to improvements. We will discuss these in the paper.
>
> **1.2 ‘do not show "clean" accuracy'; 'do not see a clear comparison or tradeoff’; ‘the results on a weak classifier**
>
> We apologize for the confusion but we do include the clean accuracy, labeled ‘ baseline clean accuracy’ in Figure 5,6,7 and we discuss it in Section 5.2 and Table 2.
>
> Regarding the ‘low’ accuracy, it attributes to the task difficulty rather than ‘a weak classifier’.
> First, for your reference, we provide the performance obtained by others on an **easier** version of this dataset. In [1], they only keep the 11 categories that appear in all years out of a total of 41 categories, and the best accuracies among the 11 methods evaluated are about 70% (Table 2 in [1]). While this is not directly comparable to our numbers because we include all 41 categories and the evaluation protocols vary, it supports the claim that the News Category dataset is challenging.
>
> Why is this challenging? It can be the distribution shifts, which is also partially why the trade-off is different from other data aggregations. Here we have the accuracy of the based learner trained on the latest n months evaluated on individual categories (also included in another rebuttal):
>
> | Category | n=1 | n=3 | n=6 | n=9 | n=12 |
> |:----------:|:------:|:------:|:------:|:------:|:------:|
> | WORLD NEWS | 59.00% | 46.59% | 36.79% | 31.39% | 25.56% |
> | POLITICS | 62.13% | 57.52% | 55.36% | 54.31% | 53.96% |
> |   |  |  |  |  |  |
> | TRAVEL | 52.30% | 56.65% | 59.46% | 61.63% | 62.26% |
> | COLLEGE | 42.30% | 49.30% | 51.68% | 54.20% | 57.28% |
>
> Here we see that for categories WORLD NEWS/POLITICS, training on earlier data reduces accuracy, indicating distribution drifts; for categories TRAVEL/COLLEGE, training on earlier data improves accuracy, where the increase of data contributes primarily. The diversity among different categories also increases the difficulty for learners.
>
> In addition, since the category of this dataset is in fact where the news is published, there can be more than one proper category, which also increases the difficulty.
>
> [1] Yao et al. Wild-time: A benchmark of in-the-wild distribution shift over time. https://arxiv.org/abs/2211.14238
>
> **2. Experiments do not verify all the claims: ‘against unbounded amounts of poisoned samples’; ‘what do you mean by provably robust’; ‘do not examine the defense against any real attacks’;…**
>
> In our threat model, we introduce earliness and duration to measure attack budgets, which are parallel to the number of poisoned samples. As we proved in Section 4.2, the baseline defense is provably robust (i.e. its prediction will not change) against poisoning attacks as long as the attacks’ earliness/duration is bounded by certain amounts. Notably, when the earliness/duration of the attack is bounded, while the number of total poisoned samples can be arbitrarily large, they cannot cover the entire timeline and therefore it won’t be the case where all samples are poisoned.
>
> The baseline defense computes the minimal attack budgets (earliness and duration in this case) that **any** attack needs to change the prediction on an input. For any given budget (earliness/duration), the defense is provably robust on a sample if the above threshold is larger than the budget and the prediction is correct (i.e. the prediction will be correct and unchanged after any attack within this earliness/duration, regardless of how many samples are poisoned, **supporting the ‘unbounded’ claim**).
>
> Following existing work with certified defenses(e.g. [3, 11, 13, 23, 33, 34] cited in the paper), we evaluate certified accuracy/fraction (robust fraction in ours) rather than against specific attack methods.
>
> **Please let us know if you find any of the answers unclear. Looking forward to your reply!**

---

> > ### Comment · Reviewer_EzrD · 2023-08-10
> > **Thank you for the rebuttal**
> >
> > (1) For the first bullet point on **Practicality of the temporal aggregation defense**, most of my concerns are well addressed. It would be great if the authors could add these discussions to the final draft.
> >
> > One remaining concern is that the authors mention that in practice, one could use a shared pre-trained feature extractor. This approach seems to preserve the utility but also involves a trade-off on influencing the provable temporal robustness as every $f_i(x)$ would be similar to each other. In the worst case, would the pre-trained feature extractor be the main factor in temporal robustness? In other words, would this empirical design affect the validity of the theory? Also, is this design mentioned somewhere in the paper or appendix? It would be great if the authors could give a pointer thus I can check the implementation details.
> >
> > (2) For the second bullet point, I suggest the authors reword the "unbounded" claim a bit as other factors are indeed bounded.
> >
> >  Also, I understand that previous papers on certified robustness do not consider real attacks, but I am indeed curious about how these series of works, especially this work perform against real data poisoning attacks. I believe this is a key factor in bringing certified robustness to the next level by examining it under a more realistic setting. Of course, at this point, I do not expect the authors to provide more experimental results, but I would love to hear maybe an educated guess from the authors as a nice discussion.  Note that any discussion on this part would not affect my score.
> >
> > According to this round of rebuttal, considering most of my concerns are addressed, I would like to raise my score. Before doing that, I would like to hear the authors' answer regarding the first question before I determine my final score. Thank you.

---

> > > ### Author Response · Authors · 2023-08-10
> > > **Thank you for the prompt response**
> > >
> > > Thank you very much for responding promptly! We are happy to know that you find our rebuttal helpful.
> > >
> > > **(1)** First of all, we will definitely add these discussions to the paper. We think these questions of yours are insightful and can be interesting to future readers.
> > >
> > > **For the follow-up questions**: We believe there are two key messages here.
> > > >(1.1) Using a pre-trained feature extractor and training only a task head for each base classifier does not mean every classifier will be similar.
> > > >
> > > >(1.2) The theoretical guarantees of our baseline defense remain valid, as the pre-trained feature extractor can be viewed as a part of the learning algorithms for base learners and does not depend on the (potentially poisoned) dataset.
> > >
> > > (1.1) We will simply use our existing experimental results to support this argument. Here is the experimental setup of the paper (included in Section 5.1 of the paper): We use the pre-trained RoBERTa-base[1] model to embed headlines into vectors with 768 dimensions and optimize linear classification heads (FYI, we experimented with more layers as implemented in *learner.py* included in the supplementary materials, but the results are not included in the paper as more layers do not increase accuracy in this case) over *normalized* RoBERTa features. While sharing the same feature extractor, the base classifiers can perform differently as we observe from this table (also included in our previous rebuttal).
> > >
> > > | Category | n=1 | n=3 | n=6 | n=9 | n=12 |
> > > |:----------:|:------:|:------:|:------:|:------:|:------:|
> > > | WORLD NEWS | 59.00% | 46.59% | 36.79% | 31.39% | 25.56% |
> > > | POLITICS | 62.13% | 57.52% | 55.36% | 54.31% | 53.96% |
> > > |      |    |    |    |    |    |
> > > | TRAVEL  | 52.30% | 56.65% | 59.46% | 61.63% | 62.26% |
> > > | COLLEGE  | 42.30% | 49.30% | 51.68% | 54.20% | 57.28% |
> > >
> > > (1.2) Assuming the pre-trained model is not trained on the (potentially poisoned) dataset that one is currently applying our baseline defense to, it can be viewed as a part of the learning algorithm that one uses to learn base classifiers, thus all our theoretical guarantees remain unchanged for the baseline defense. However, it is worth noting that the quality of the feature extractor (or whether to use a pre-trained feature extractor) can affect the quality of the learning algorithm and therefore the performance of base classifiers, which affect the performance and robustness of the entire defense (i.e. better base classifiers can mean better defenses). Thus it also won't be accurate to claim that the provable robustness does not depend on the feature extractor.
> > >
> > > **(2)** We see how the current 'unbounded' claim can be misunderstood. We will revise it by emphasizing the condition being 'when the attacks are bounded temporally'.
> > >
> > > For aggregation defenses against empirical poisoning attacks: This is a great question. For the traditional data poisoning threat models (i.e. using the number of poisoned samples as attack budgets), DPA and Finite Aggregation ([13, 33] in our paper), among many other kinds of defenses, are evaluated by [2] as empirical defenses against various of dirty-label/clean-label backdoor attacks, and 'DPA and Finite Aggregation consistently offer very high levels of robustness for both attacks' as in Figure 1 of [2]. In addition, [3] also looks into this matter but targets more challenging scenarios to probe the limits, which you may find interesting.
> > >
> > > As for the temporal threat model, we believe different empirical attacks will vary in terms of how many samples they need to inject into different time periods, but they won't be very different in terms of temporal budgets assuming they can break each individual base classifier.
> > >
> > > **It is nice to discuss this with you. Thanks for your service!**
> > >
> > > [1] Liu et al. Roberta: A robustly optimized bert pretraining approach.(https://arxiv.org/abs/1907.11692)
> > >
> > > [2] Baracaldo et al. Benchmarking the Effect of Poisoning Defenses on the Security and Bias of the Final Model (https://openreview.net/forum?id=PP3H72O_E2f)
> > >
> > > [3] Wang, W, and Feizi, S. On Practical Aspects of Aggregation Defenses against Data Poisoning Attacks. (https://arxiv.org/abs/2306.16415)

---

> > > > ### Comment · Reviewer_EzrD · 2023-08-11
> > > >
> > > > Thank you for the reply, I have raised my score accordingly.

---

> > > > > ### Author Response · Authors · 2023-08-11
> > > > > **Thank you!**
> > > > >
> > > > > Thank you again for your services and insights!
> > > > >
> > > > > We will use these discussions to improve our work.
> > > > >
> > > > > Have a good day!

---

### Official Review · Reviewer_TaU1 · 2023-07-06

**Soundness:** 3 good
**Presentation:** 3 good
**Contribution:** 2 fair
**Rating:** 5
**Confidence:** 4

**Summary:**

The paper discusses the concept of data poisoning, wherein adversaries manipulate machine learning algorithms through the inclusion of malicious training data. The existing threat models in this domain primarily revolve around a single metric, namely the number of poisoned samples. However, if attackers can introduce a greater number of poisoned samples than expected, while incurring reasonable overhead, they can swiftly render existing defense mechanisms ineffective. In order to address this challenge, the authors propose leveraging timestamps associated with the creation dates of data, which have been overlooked in prior research. By capitalizing on these timestamps, the authors introduce a temporal threat model for data poisoning, introducing two novel metrics: earliness and duration. The earliness metric measures the time interval before an attack is initiated, while the duration metric quantifies the duration of an attack. By utilizing these metrics, the authors define the concept of temporal robustness against data poisoning, which offers meaningful protection even in scenarios with an unrestricted number of poisoned samples. To facilitate empirical evaluation of temporal robustness, the authors present a benchmark with an evaluation protocol that emulates continuous data collection and periodic deployment of updated models. Finally, the authors propose and empirically verify a baseline defense mechanism called temporal aggregation, which provides verifiable temporal robustness and underscores the potential of the proposed temporal threat model in combating data poisoning.

**Strengths:**

+ An interesting and novel temporal threat model for data poisoning is formalized and proposed.
+ Two new metrics, earlyness and duration, are designed to represent the temporality of data poisoning.
+ Tailor the concept of temporal robustness to data poisoning and develop and empirically validate a baseline defense, temporal aggregation.
+ Provides a comprehensive benchmark evaluation for temporal data poisoning.

**Weaknesses:**

- The practicality of the threat model requires further evaluation.
- Evaluations are still incomplete in terms of attack strategies and benchmark datasets.
- The rationality of the attack scenario still needs further clarification.
- The novelty of the designed defense strategy still needs to be further strengthened.

**Questions:**

Please kindly refer to the comments in limitations.

**Limitations:**

- The practicality of the threat model requires further evaluation.
     - First, this paper emphasizes temporal data poisoning and considers long-term poisoning attacks on ML models by adversaries within reasonable overhead. However, it would be helpful to provide a bit more context on the practical scenarios in which affordable overhead allows attackers to poison more samples than expected. Elaborating on such scenarios would enhance the understanding of the problem's significance.
    - Second, the authors need to further elaborate on the background of the attacker, the knowledge of the attacker, the limitations of the attacker, and the capabilities of the attacker.
    - Last, the reviewer is a bit curious about how an adversary can achieve continuous poisoning data injection without the timestamp being disturbed by other data sources. For example, when the adversary is injecting poisoned data according to the timestamp, other benign users are also inserting data. In this case, can it be understood that the poisoned data has been diluted? If the above assumptions are true, such an attack seems to be difficult to pose a substantial threat to the ML model.

-  Evaluations are still incomplete in terms of attack strategies and benchmark datasets. This paper uses the News Category Dataset to evaluate the robustness of temporal data poisoning. Such an approach has the following limitations:
    - First, in order to more fully evaluate the robustness of temporal data poisoning, the authors should add more benchmark datasets to verify the authors' claims.
    - Second, the reviewers are more curious about whether the time series data set can better evaluate temporal data poisoning, such as IoT time series data, financial time series data, etc.
    - Last, did the authors consider extreme cases such as continuous and long-term poisoning data injection by an adversary? The reason is that this can experimentally illustrate the bounds of the robustness of temporal data poisoning.

- The novelty of the designed defense strategy still needs to be further strengthened. Although the temporal aggregation proposed in this paper is a baseline solution, there are still the following problems to be solved:
    - First, in order to more comprehensively verify the provable temporal robustness of temporal data poisoning, the authors should consider multiple naive schemes, such as validation set testing, ensemble learning, weighted aggregation, etc.
    - Second, the paper still lacks an evaluation of edge cases, which is helpful to understand the limits of defense schemes.

---

> ### Author Rebuttal · Authors · 2023-08-08
>
> Thanks for reviewing. We are glad that you find the threat model interesting. Meanwhile, there may be misunderstandings since our work does **NOT** involve *making poisoning attacks long-term so that there will be more and more poisoned samples over time to break defenses*. Thus we want to recap our contributions briefly.
>
> >**Recap of contributions**
> >
> >
> >The purpose of this work is to propose a **new threat model** for defenses against data poisoning. Most existing threat models use the number of poisoned samples for attack budgets. What we do is measure the attacker’s effort by **how early when the attack started** (earliness) and **how long the attack lasted** (duration), so that we have other tools when the number of poisoned samples is insufficient to characterize the efforts of the attackers.
> >
> >
> >Then we define robustness notions similarly to existing threat models by replacing attack budgets with upper bounds for earliness and duration. We design a benchmark for empirical evaluations and propose a baseline defense to show that such robustness is achievable.
> >
> >
> >**In short**, we are proposing a new threat model for cases that are particularly challenging to the existing ones, with its usability supported by both an empirical benchmark and a certifiably robust baseline defense.
>
> We believe some of your questions should be naturally resolved with this clarification, and we will address the others for you.
>
> **1. practicality of the threat model**
>
> **1.1. the significance**
>
> The need for new metrics of attack budgets originates from the difficulty of defending against poisoning attacks relying on the old metric, i.e. the number of poisoned samples. The issue here is that the numbers of poisoned samples we can tolerate are still limited that requiring attackers to poison more samples than we can tolerate may not enforce much cost to them.
> This is supported both theoretically [1] and empirically [2, 3, 4] by existing work.
>
> [1] Wang et.al., Lethal Dose Conjecture on Data Poisoning.
>
> [2] Souri et.al., Sleeper Agent: Scalable Hidden Trigger Backdoors for Neural Networks Trained from Scratch.
>
> [3] Geiping et.al., Witches' Brew: Industrial Scale Data Poisoning via Gradient Matching.
>
> [4] Carlini et.al., Poisoning Web-Scale Training Datasets is Practical.
>
> **1.2. on the knowledge/capabilities of the attacker**
>
> As we mentioned above in the essence of our work, we are proposing new metrics of attack budgets for defenses of data poisoning. Thus regarding assumptions on the attacker’s knowledge and capabilities, the proposed threat model is in fact compatible with most (if not all) settings from existing data poisoning threat models, i.e. one can also consider attackers with varying knowledge of the defenses/tasks/data and varying resources.
>
> For the baseline defense, as we prove in Section 4, it is robust to all attacks with bounded earliness/duration, regardless of the knowledge/resources of the attackers.
>
> **2. 'Evaluations are still incomplete'**
>
> **2.1 attack strategies and benchmarks**
>
> Recall in our recap that the main goal of the baseline defense is to show that the temporal robustness notions can be provided. Since it is a certified defense, one can compute the minimal attack budgets (earliness and duration in this case) for **any** attack to have a chance of breaking it. Thus following existing work with certified defenses(e.g. [3, 11, 13, 23, 33, 34] cited in the paper), we evaluate their certified accuracy/fraction (robust fraction in our paper) rather than against specific attack methods.
>
> Regarding benchmarks, since this is a brand new threat model for data poisoning, there was no existing benchmark available. We construct a benchmark for empirical evaluations from the News Category Dataset due to both its availability and its natural distribution shifts over time. We believe our benchmark is a good start for research purposes but we will look into ways to construct better benchmarks in future work.
>
> **2.2 whether the time series data set can better evaluate temporal data poisoning**
>
> Although the time series data naturally involves temporal concepts, they are different from the timestamps we incorporate here in the temporal threat model of data poisoning (i.e. timestamps denoting the birth dates of individual samples). Thus while the temporal threat model applies to tasks with time series data if the data collection process can associate each sample with its birth date, existing time series datasets are not necessarily better candidates for research benchmarks than what we have already. We would like to hear your insight if possible.
>
> **3. novelty of the designed defense strategy**
>
> We argue that the novelty of the baseline defense is not the key contribution of this work, in fact, the technique itself is primarily adapted from aggregation defenses in the existing threat model of data poisoning, with minimal modifications. The major novelty of this work is from the novel threat model of data poisoning that utilizes temporal concepts, and the simple baseline defense supports the threat model by showing that (provable) temporal robustness is achievable.
>
> **3.1 'should consider multiple naive schemes, such as validation set testing, ensemble learning, weighted aggregation, etc.'**
>
> We are not sure how to interpret these terms as naive schemes for temporal poisoning robustness. Could you please elaborate more?
>
> **3.2 'lacks an evaluation of edge cases'**
>
> As clarified earlier, since the baseline defense is provable, one can compute the minimal attack budgets (earliness and duration in this case) for **any** attack to have a chance of breaking it. Thus following existing work with certified defenses(e.g. [3, 11, 13, 23, 33, 34] cited in the paper), we evaluate certified accuracy/fraction (robust fraction in our paper) rather than against specific attack methods.
>
> **Thanks again for your review.**
>
> **Please let us know if any of your questions/concerns remain after rebuttal.**

---

> > ### Comment · Reviewer_TaU1 · 2023-08-14
> > **Response to Authors' rebuttal**
> >
> > Thanks to the authors for their detailed rebuttals! Most of my concerns have been addressed. I will maintain the current rating.

---

> > ### Comment · Reviewer_TaU1 · 2023-08-21
> > **Response to Authors' rebuttal**
> >
> > After much thought and confirmation, I will revise my rating. Thanks again to the authors for their detailed responses.

---

### Official Review · Reviewer_NyPv · 2023-07-07

**Soundness:** 3 good
**Presentation:** 4 excellent
**Contribution:** 3 good
**Rating:** 5
**Confidence:** 4

**Summary:**

The paper considers "temporal robustness", an extension to traditional data poisoning threat models which considers the "birthdate" of training examples. This extension gives more power to a defender, by permitting defenses against adversaries limited in the duration or earliness of their attack. The paper proposes a simple defense, temporal aggregation, which provides provable temporal robustness, and evaluates it on a benchmark derived from the News Category dataset.

I appreciate the authors' response and will keep my score.

**Strengths:**

The paper proposes an interesting, well thought out threat model. I enjoyed reading section 2. One comment is that it is unclear when assumption 4.2 is used. Are there other versions of Section 4.2 that might be possible in different applications, or does modifying the assumption in any way prevent any guarantees?

Temporal aggregation is a natural defense here. Figure 4 is a nice description of the technique.

The threat model seems nicely amenable to the continual training setting, and I'd enjoy reading followup applying it there.

**Weaknesses:**

There is only one dataset. I appreciate that the setting necessitates constructing new benchmarks, but the ultimate benchmark here seems to be not enough to promote future research. I would encourage the authors to think about what properties of a temporal robustness benchmark could make defenses challenging, or more straightforward. After thinking briefly, some that come to mind are seasonality in the data, and distribution drift. Larger amounts of both seem like they could make attacks easier, and it would be interesting to evaluate benchmarks with differing levels of both properties. Right now, it doesn't appear that either property is present, given the very small amount of performance degradation when training on other months' data.

The paper provides limited motivation about temporally bounded adversaries. While the idea of birthdates is very natural, and their existence is well motivated in the paper, it's not very clear from the paper why an adversary would have a bounded duration. I can imagine more settings with bounded earliness, but the paper should still include examples. This also is reflected in the benchmark, which doesn't really justify why this setting might run into temporal robustness.

Temporal aggregation seems to be overkill for a bounded earliness adversary. Why not just train on all but the last tau sections of the dataset? Then you wouldn't need the min(tau, k) term.

Comments:

The Antidote paper [x1] considers temporally bounded adversaries. There is a recent paper [x2] on poisoning web datasets which considers temporally bounded adversaries which may be of interest.

Please don't use [xx] to cite, prefer e.g. "Jia et al. [11] do xxx" to "[11] do xxx".

Assumption 2.4 is used in Definition 2.3, before the assumption is introduced.

[x1] - https://dl.acm.org/doi/pdf/10.1145/1644893.1644895

[x2] - https://arxiv.org/abs/2302.10149

**Questions:**

I've left multiple questions in the strengths and weaknesses section. Here's a summary:

1. Can bounded earliness defenses be simpler by never considering the samples after the earliness threshold?

2. When is assumption 4.2 needed? Are there any modifications of it that would also make sense?

**Limitations:**

There isn't really discussion of limitations in the paper.

---

> ### Author Rebuttal · Authors · 2023-08-07
>
> Thank you! We are glad that you find our threat model interesting and well thought out. Our answers to your questions are as follows and **we will incorporate them in our paper as they can be interesting to future readers as well**.
>
> **1. distribution drift in the benchmark:**
>
> The benchmark we presented is in fact rich in distribution drifts. In the paper, we presented in Figure 3 the changes of category presences in different months as support, and we add more evidence here to explain why ‘*the very small amount of performance degradation when training on other months' data*’ does not mean missing distribution drifts.
>
> Here we have the accuracy of the based learner trained on the latest n months with varying n, but evaluated on individual categories:
>
> | Category | n=1 | n=3 | n=6 | n=9 | n=12 |
> |:----------:|:------:|:------:|:------:|:------:|:------:|
> | WORLD NEWS | 59.00% | 46.59% | 36.79% | 31.39% | 25.56% |
> | POLITICS | 62.13% | 57.52% | 55.36% | 54.31% | 53.96% |
> |      |    |    |    |    |    |
> | TRAVEL  | 52.30% | 56.65% | 59.46% | 61.63% | 62.26% |
> | COLLEGE  | 42.30% | 49.30% | 51.68% | 54.20% | 57.28% |
>
> Here we see that for categories WORLD NEWS/POLITICS, training on earlier data reduces accuracy, indicating distribution drifts; for categories TRAVEL/COLLEGE, training on earlier data improves accuracy, where the increase of data contributes primarily. Thus the mild degradation of accuracy in Table 2 is a joint effect of both natural distribution drifts and varying numbers of data (i.e. data quality v.s. data quantity).
>
> We will include these (and other) per-class accuracies and the discussions in our paper to better illustrate the distribution shifts in our benchmark. Thank you for bringing it up!
>
>
> **2. motivations about temporally bounded adversaries**
>
> Conceptually, the main motivation for considering temporally bounded adversaries is that time is an important and limited resource, thus as humans, assuming the same rewards, our instincts would prefer instant rewards over rewards that require waiting and would prefer rewards that require less time of work than rewards that require more time of work, especially if the scales come to months or years.
>
> In the case of poisoning, for attackers, the easier target is probably a system that can be broken soon rather than a system that won’t be broken for at least another several months (hopefully years with more progress in temporal defenses). This motivates the use of earliness.
>
> Meanwhile, if the target system is not robust to attacks that end promptly, one can poison enough samples shortly and then just wait. But if the defense is robust to attacks with bounded durations, the attacker would have to put effort for a longer time range (e.g. poison some samples for more months). We believe ‘duration’ characterizes such preference.
>
> In addition, while the threat model is new, existing attacks usually are bounded temporally since it always takes time to make attacks earlier or longer (no shortcut). In fact, the additional references [x1, x2] you provided that involve temporally bounded attackers can be helpful as a background. We will discuss them in the paper.
>
> For the benchmark, we select the News Category Dataset mostly due to its natural distribution shifts over time. Classifying the categories of news is indeed not typically considered a critical task that can result in serious damages after being attacked. However, since similar situations apply to other benchmarks used in data poisoning such as MNIST, CIFAR-10, and GTSRB, we believe our benchmark is a good start for research purposes. We will look into ways to construct better benchmarks in future work.
>
> **3. Can bounded earliness defenses be simpler by never considering the samples after the earliness threshold?**
>
> The goal of the temporal threat model is to utilize temporal concepts to raise the attack cost (i.e. time spent on poisoning and waiting). While ignoring the latest samples is a simpler defense for poisoning attacks with bounded earliness, it is potentially vulnerable to attacks with small durations, i.e. an attack may inject enough poisoning samples shortly and wait for the defense to be broken.
>
> On the other hand, the baseline defense we have in our paper is in fact a single solution (and we believe it is fairly simple) serving as provable defenses with respect to two budgets, earliness and duration, i.e. an attack must have both sufficient earliness (starts early enough) and sufficient duration (lasts long enough) in order to have a chance of breaking the defense.
>
> In addition, our baseline defense is also a provable defense with a bounded number of poisoned samples when using small n, which follows directly from previous aggregation defenses like DPA [13] for existing poisoning threat models. This is an additional advantage compared to simply ignoring the latest samples.
>
> [13] Alexander Levine and Soheil Feizi. Deep partition aggregation: Provable defenses against general poisoning attacks. ICLR 2021
>
> **4. Regarding assumption 2.4**
>
> In assumption 2.4, we assume the attacker has no capability to directly set the birth dates of samples. The role of assumption 2.4 is to connect the temporal metrics of attacks (i.e. earliness and duration) with the actual cost of attackers: If an attacker wants to increase the earliness or duration of its attack, it has to actually start early or poison for a longer period.
>
> While we believe assumption 2.4 is natural and practical, it is not the only working assumption. For example, the threat model remains meaningful if the attacker can set the timestamps but there are some constraints on the values (e.g. it cannot be too far away from the actual time).
>
> **For your comments:** Thank you for providing additional references and suggestions for paper formatting. We will include them in our paper.
>
> **Thanks again for your services. We look forward to hearing your comments on our answers and welcome any follow-up.**

---

> > ### Author Response · Authors · 2023-08-21
> >
> > Dear Reviewer NyPv,
> >
> > Thank you again for the comments and suggestions in your review.
> > Since today is the last day of author-reviewer discussion, we would kindly ask you to check our rebuttal and let us know if you have any further questions.
> >
> > Again, thank you for your time!

---

### Decision · Program_Chairs · 2023-09-21

**Decision:**

Accept (poster)

**Comment:**

The paper considers a novel angle to study poisoning attacks and defenses by considering the temporal dimension and defining two corresponding metrics other than just considering the fraction of poisoning samples the attacker can control. The rebuttal has clarified the major issues raised by the reviewers; however, the threat model is still somewhat unconvincing. Thus, the authors should clarify it and provide additional motivation behind it when preparing the final version.